The real-time hand and object recognition for virtual interaction

Nuralin Madi 1
Daineko Yevgeniya 2
Aljawarneh Shadi 3
Tsoy Dana d.tsoy@iitu.edu.kz 1
Ipalakova Madina 4
1 Mixed Reality Laboratory, International Information Technology University , Almaty , Kazakhstan
2 Radio Engineering, Electronics and Telecommunication Department, International Information Technology University , Almaty , Kazakhstan
3 Faculty of Computer and Information Technology, Jordan University of Science and Technology , Irbid , Jordan
4 Computer Engineering Department, International Information Technology University , Almaty , Kazakhstan
Coelho Paulo Jorge
Electronic publication date: 2024 Jun 27
Publication date: 2024
Volume: 10
Electronic Location ID: e2110
Received 2023 Dec 21; Accepted 2024 May 16
Copyright: ©2024 Nuralin et al.
Copyright year: 2024
Copyright holder: Nuralin et al.
License: This is an open access article distributed under the terms of the Creative Commons Attribution License, which permits unrestricted use, distribution, reproduction and adaptation in any medium and for any purpose provided that it is properly attributed. For attribution, the original author(s), title, publication source (PeerJ Computer Science) and either DOI or URL of the article must be cited.
License URL: https://creativecommons.org/licenses/by/4.0/

Keywords: Mediapipe, Hand-object interactions, Virtual interactions, EXtended reality (XR), Computer vision

Funding: Science Committee of the Ministry of Education and Science of the Republic of Kazakhstan Grant No. AP14871641 This research was funded by the Science Committee of the Ministry of Education and Science of the Republic of Kazakhstan (Grant No. AP14871641). The funders had no role in study design, data collection and analysis, decision to publish, or preparation of the manuscript.

==============================
Recognizing hand-object interactions presents a significant challenge in computer vision. It arises due to the varying nature of hand-object interactions. Moreover, estimating the 3D position of a hand from a single frame can be problematic, especially when the hand obstructs the view of the object from the observer’s perspective. In this article, we present a novel approach to recognizing objects and facilitating virtual interactions, using a steering wheel as an illustrative example. We propose a real-time solution for identifying hand-object interactions in eXtended reality (XR) environments. Our approach relies on data captured by a single RGB camera during a manipulation scenario involving a steering wheel. Our model pipeline consists of three key components: (a) a hand landmark detector based on the MediaPipe cross-platform hand tracking solution; (b) a three-spoke steering wheel model tracker implemented using the faster region-based convolutional neural network (Faster R-CNN) architecture; and (c) a gesture recognition module designed to analyze interactions between the hand and the steering wheel. This approach not only offers a realistic experience of interacting with steering-based mechanisms but also contributes to reducing emissions in the real-world environment. Our experimental results demonstrate the natural interaction between physical objects in virtual environments, showcasing precision and stability in our system.

Introduction

In recent years, the rapid advancement of immersive technologies has revolutionized the way humans interact with virtual environments. A fundamental aspect enhancing this interaction is real-time hand and object recognition, crucial for fostering natural and intuitive information exchange.

The development of virtual and augmented realities has expanded the scope of virtual interaction beyond traditional human–computer interfaces, enabling users to interact with digital content in unique ways. Accurate recognition and interpretation of hand gestures and physical objects in real time are central to this evolution. Whether manipulating virtual objects, navigating virtual environments, or engaging in virtual collaboration, effective and seamless interactions rely heavily on precise and speedy hand and object recognition systems.

Robust real-time hand and object recognition algorithms have emerged from advancements in computer vision, machine learning, and sensor technologies. These algorithms, employing complex neural networks and deep learning techniques, analyze visual data in real time, enabling precise tracking of hand movements and accurate identification of various objects.

Tracking hand movements and gestures facilitates a more natural interaction between hands and objects within virtual reality. It empowers the creation of interfaces and applications that require minimal user effort, eliminating the need to memorize specific movements or instructions. The concept of natural interaction embodies intuitive engagement.

Moreover, integrating hand and object recognition into virtual interaction reveals numerous practical applications across various domains, including immersive gaming experiences, virtual learning simulations, interactive educational platforms, and telepresence systems. The potential impact of real-time recognition technologies is vast and continuously evolving.

However, despite significant progress, challenges persist in hand and object recognition for real-time virtual interaction, such as occlusion handling, resilience to environmental changes, technology accessibility, scalability across diverse hardware platforms, and concerns regarding privacy and security in data processing.

Given these considerations, the aim of this article is to create a simple yet cost-effective solution for interacting with objects in virtual reality using minimal special devices and applications. By addressing these challenges and harnessing the full potential of advanced recognition technologies, we aim to enhance the immersive nature of virtual environments and redefine the boundaries of human–computer interaction.

The authors proposed an accessible method for real-time interaction between hands and objects within extended reality (XR) systems. The system comprises three components: a hand landmark detector, a steering wheel model tracker, and a gesture recognition module designed to analyze the interaction between the hand and the steering wheel. Experimental results demonstrate that the system accurately reproduces interactions with physical objects in virtual environments.

Related Works

The fourth wave of industrialization has accelerated the rapid development and integration of various technological tools into our daily lives, significantly impacting the demand for enhanced interactions between users and digital products and devices. This surge in technology has led to the evolution of new approaches in human-computer interaction (HCI). Furthermore, with the widespread adoption of artificial intelligence across various industries, HCI has also undergone significant transformations.

While traditional input systems remain prevalent, there is a noticeable shift towards interfaces that reduce user effort. One such example is hand-object interaction tracking.

In recent decades, extensive research has been conducted in the field of action recognition using deep learning, resulting in the creation of numerous deep learning models and datasets for training and testing purposes. Notable datasets include HMDB51, Sports1M, YouTube8, Activity, among others. A comprehensive overview of the development history of deep learning in hand-object interaction recognition can be found in Zhu et al. (2020), where the authors analyze over 200 projects, methods, and models in this field.

The approaches mentioned above for utilizing specific equipment and cameras vary depending on the application. For instance, virtual reality (VR) often relies on a variety of controllers, such as joysticks, gamepads, VR gloves, and other similar devices. This emphasis on user input is crucial as VR operates in a virtual environment largely isolated from the real world, making user input the primary consideration.

Conversely, in the context of augmented reality (AR) and mixed reality (MR), where virtual objects are integrated into the real environment alongside the user, it becomes essential to account for all physical objects. To tackle this challenge, Microsoft Kinect sensors and cameras are frequently employed as they enable observation not only of the user but also of the physical surroundings. Some systems, such as the Microsoft HoloLens 2 and Oculus Quest 2, utilize built-in cameras capable of tracking the user’s hands.

In this regard, Sakamoto et al. (2020) conducted a study focusing on user interactions with virtual objects through fundamental manipulations, including creating, deforming, moving, and combining these virtual entities. To facilitate these interactions, they utilized web cameras, the Leap Motion Controller, and a computer. Overall, their approach garnered positive feedback, as reflected in questionnaire responses. Users expressed that the manipulation method was intuitive and easy to grasp, noting that their hand movements were accurately reflected in the system.

Some works demonstrate innovative approaches to interaction within virtual environments, significantly reducing user effort. Rehman, Ullah & Khan (2023) propose four techniques that rely on the pose of a single fingertip. It makes interaction efficient requiring minimal learning effort for users.

This research focuses on a real-time hand gesture recognition system designed for dynamic applications. Utilizing a webcam, the system captures hand movements and translates them into specific commands. The authors emphasize the significance of real-time processing and user error tolerance in such systems (Rautaray, 2012).

In their study, Kim, Kim & Yoo (2013) describe the development of a hand-gesture-based user interface designed for interacting with virtual objects in three-dimensional (3D) environments. The system enhances the accuracy and flexibility of virtual object control by employing various filtering operations on sequences of hand-gesture images captured by the Kinect sensor. Hand gesture events are processed using callback and delegate functions.

In their research, Wu et al. (2020) utilized Hand Pose Estimation to predict the positions of hand joints from images. They developed a system capable of estimating hand poses in 3D space using depth images, primarily for VR applications. Their approach, relying on convolutional neural networks (CNN), effectively addressed the occlusion problem and could predict joint positions even in complex scenarios, including bare hands and hands manipulating various objects.

These studies highlight the importance of recognizing hand manipulation actions within XR applications. Such recognition enhances interaction by making it more intuitive and natural, with high performance. However, it is worth noting that these approaches typically require additional equipment, such as specialized controllers or sensors.

An alternative solution, as suggested, is the utilization of cameras, which has the potential to eliminate the need for extra hardware, offering a more accessible and convenient approach to XR interaction.

Thus, cameras enable the capturing and processing of images from the surrounding environment to define interactions. The advent of wearable camera devices has facilitated the study of human and object interaction from a first-person perspective, known as egocentric action recognition. This advancement has spurred numerous research efforts and the development of additional datasets (Damen et al., 2020a; Damen et al., 2020b).

Furthermore, as the use of extra devices decreases, additional gestures are gradually being removed from virtual environment interaction (Raees et al., 2021). The authors propose the thumb inclination method of manipulation within created media.

In the realm of egocentric action recognition, there is a special emphasis on comprehending the dynamics of human hands and their interaction with objects. Kwon et al. (2021) have highlighted certain limitations in existing datasets and studies, particularly the lack of a comprehensive understanding of actions in 3D.

The recordings captured by egocentric cameras are rich in complex information, including object tracking and hand pose estimation. However, these videos are susceptible to challenges like background clutter and occlusion due to the constantly changing perspective of the camera wearer.

To tackle these issues, various approaches have been proposed. One notable method is the dense trajectories approach, which incorporates head motion as a form of motion compensation (Li, Ye & Rehg, 2015). This technique helps mitigate the impact of background clutter and occlusion, enhancing the robustness of egocentric video analysis.

Another challenge revolves around the limitations of existing action recognition datasets. Most of them provide data for abstract objects represented as cuboids (Sridhar et al., 2016; Garcia-Hernando et al., 2018; Hampali et al., 2020; Hasson et al., 2019). This issue arises because the human hand can perform entirely different actions with different types of objects. Therefore, it is crucial to approach action recognition in terms of object classes. Kwon et al. (2021) proposed a method for recognizing first-person two-hand interactions. The authors developed the Two Hands Object Manipulation (H2O) dataset, which facilitates pose prediction and has demonstrated promising results.

Several articles incorporate object detection features to predict the precise context of an object, aiming to enhance egocentric video recognition (Wang et al., 2020; Singh et al., 2016; Sener, Singhania & Yao, 2020). However, some studies focus solely on single-hand manipulation cases (Sridhar et al., 2016; Garcia-Hernando et al., 2018; Hampali et al., 2020), while in certain scenarios, hand manipulation involves both hands (Kwon et al., 2021).

Given the expansive scope and complexity of the subject matter, as well as the existing methodologies and their limitations, this study delves into a specific case: the recognition of steering wheel manipulation. Its practical application is particularly valuable in autopilot systems, where hand analysis enables the prediction of preparatory movements for maneuvers. Analyzing the driving process carries the potential to mitigate the consequences of distracted driving, a common cause of road accidents (Craye & Karray, 2015). Moreover, it holds relevance in military simulators for refining manipulation skills across diverse types of machinery.

Finally, hand tracking and its interaction with the vehicle control system represent significant steps toward the transition to natural user interfaces (Hepperle et al., 2019), enabling drivers to control the vehicle using only hand gestures. Borghi et al. (2018) have introduced a new dataset for hand recognition and tracking in driving conditions, simplifying the prediction of drivers’ behavior, and fostering a more user-friendly and secure human-car interaction. Given these challenges, this article outlines its approach as follows:

− development of an occlusion-free hand tracking and pose estimation module;

− implementation of a steering wheel tracking solution.

− analysis of the interaction between the hand and the steering wheel within the context of driving.

While the proposed methods offer innovative approaches to virtual reality interaction using minimal tools, real-time gesture recognition still demands significant computational resources and powerful algorithms.

Hand Pose Estimation

Pose estimation can be achieved through various tools and methods, as discussed earlier. One method involves hand pose estimation, where heatmap images are analyzed using CNN architecture regression (Iqbal et al., 2018). Ge et al. (2019) proposed reconstructing the complete 3D mesh of the hand’s surface.

To address video frame errors such as occlusion, low resolution, and noise, a model-based approach utilizing the MANO model was employed to capture hand pose and shape (Romero, Tzionas & Black, 2022). Subsequently, several enhancements in the model-based approach were implemented through linear interpolation calculations for the surface mesh of the hand model (Hasson et al., 2019). Thus, Boukhayma, Bem & Torr (2019) suggest compiling synthetic data for model pre-training to improve the accuracy of hand pose estimation.

A unique approach to real-time hand tracking and pose estimation was developed by a team of researchers at Google AI. Integrated into the MediaPipe framework, this solution simplifies 2.5D landmark predictions and relies on basic equipment, making the process nearly effortless (Zhang et al., 2020). The approach involves two models working together:

(1) the palm detector, also known as BlazePalm, which takes a full image from a video frame and returns an oriented bounding box;

(2) the hand landmark model, which identifies 21 key points on the hand in 3D space, enabling the reconstruction of the hand skeleton (Fig. 1).

Figure 1 MediaPipe Hand landmarks.

Interaction Recognition

Interaction recognition seeks to understand how objects and hands interact. Two main approaches are typically used: tracking hand poses based on predictive modeling (Ge et al., 2019; Moon, Chang & Lee, 2018; Mueller et al., 2017; Oberweger & Lepetit, 2017; Simon et al., 2017) or tracking objects independently (Brachmann et al., 2016; Li et al., 2018; Peng et al., 2019; Tekin, Sinha & Fua, 2018).

However, action recognition from a first-person perspective presents several distinct challenges such as rapid camera movement, extensive occlusions, and background clutter (Li, Ye & Rehg, 2015). Finally, a limitation in previous research is the lack of exploration into the reasoning behind hand and steering wheel actions.

The article focuses on developing a model for interactions between real objects and a virtual environment, using a steering wheel as an example. An RGB camera is utilized for gesture tracking, enabling real-time object recognition. The proposed method demonstrates that achieving intuitive and efficient interaction within virtual reality can be both accessible and straightforward in terms of equipment.

Materials & Methods

This section outlines the equipment and the pipeline for the project, aiming to create an affordable, straightforward, yet efficient system for recognizing hand-object interactions. The equipment setup consists of basic devices, including the Fanatec Porsche 911 GT2 gaming wheel, Acer C24 monitor, and an integrated RGB web camera. This toolset is based on the existing equipment of the research laboratory at the university. Figure 2 provides an overview of the proposed pipeline, where the model captures synchronized video frames from a single RGB camera.

Figure 2 The proposed hand-steering wheel action recognition model.

To detect the initial steering wheel location, we utilize a faster region-based convolutional neural network (Faster R-CNN). This choice was made due to its suitability for detecting specific areas of interaction and its flexibility for training. The implementation of the architecture involves the following steps:

(1) Extraction of a feature map from an image: The resulting feature map with dimensions cH/6 W/6 is generated through the application of a 3 ×3 convolutional layer. Padding is set to one to ensure that the final matrix retains its size. Each cell (i, j) of the feature map corresponds to a vector of dimension c.

(2) Generation of hypotheses based on the map obtained of object signs (Region Proposal Network): This step involves determining approximate coordinates and identifying the presence of an object belonging to the steering wheel class.

(3) Matching the coordinates of the hypotheses using regions of interest (RoI) with the feature map obtained in the first step.

(4) Classification of hypotheses, specifically for identifying the steering wheel class, and optional refinement of coordinates (though this step may not always be applied).

Hand Tracking

The MediaPipe hand landmark architecture incorporates four input parameters: real-world images, synthetic images, hand presence, and handedness hypothesis.

Upon palm detection across the entire image, the model generates a consistent internal hand pose representation, producing three outputs: (a) 21 hand landmarks, including relative depth, x, and y coordinates, as illustrated in Fig. 3; (b) an indicator estimating the presence of a hand in the given image; (c) binary classification for determining handedness, distinguishing between right and left hands, for example.

Figure 3 MediaPipe Hand Landmark detection architecture.

By supplying the precisely cropped palm image to the hand landmark model, the necessity for data augmentation, such as rotations, translations, and scaling, is significantly reduced. This approach allows the network to focus most of its capacity on enhancing landmark localization accuracy.

In a real-time tracking scenario, a bounding box is extrapolated from the landmark prediction of the previous frame, which serves as input for the current frame. This approach eliminates the need to repeatedly apply the detector on every frame. Instead, the detector is utilized only on the initial frame or when the hand prediction indicates that the hand has been lost.

Steering Wheel Detector

The search for the steering wheel location relies on the model depicted in Fig. 4, which is based on key point detection. Due to the absence of dedicated steering wheel datasets, we collected 112 images specifically for this task. These images encompass various situational poses both inside and outside the car. To align with the RCNN architecture, we utilized the YoloV5 method for annotating each image. Each annotated image in the dataset contains the following information:

Figure 4 Three-spoke steering wheel model.

− bounding box location: each object is enclosed within a bounding box, defined by its bottom-right and top-left corners in [x1, y1, x2, y2 ] format, which serves as an object segmentation tool within the image;

− key point coordinates: in the proposed model, a total of 4 key points is identified and represented in the format [x, y, visibility].

Algorithm 1 describes the setup for steering wheel detection step by step.

	
Algorithm 1	
1 # Initialize the parameters	
2 confThreshold := 0.5	
3 maskThreshold := 0.3	
4 # Load the model	
5 weights_path #Pre trained model weights	
6 # Initialize the video stream	
7 cv2.VideoCapture(path_to_video)	
8 Process frames	
9 frame: = cv2.read()	
10 blob: = cv2.dnn.blobFromImage(frame)	
11 convNet.set(blob)	
	
12 # Retrieve the bounding boxes and draw the box for each detected object	
13 for in range(num_detections):	
14 for bbox in bboxes:	
15 start_point = (bbox[0], bbox[1])	
16 end_point = (bbox[2], bbox[3])	
17 image = cv2.rectangle(image.copy(), start_point, end_point, (0,255,0), 2)	
18 for kps in keypoints:	
19 for idx, kp in enumerate(kps):	
             a. image = cv2.circle(image.copy(), tuple(kp), 5, (255,0,0), 10)	
	

In steps 1-5, the algorithm initializes. Step 8 involves loading a video stream. Steps 10-12 involve retrieving frames from the video stream. In step 13, the frame blob is applied to the CNN, as discussed in Section 2. Finally, steps 15-23 visualize bounding boxes and key points based on detections.

Steering Wheel Model

After the steering wheel detection, the three-spoked model accurately determines the steering wheel coordinates within the detected bounding box using regression. This model consists of distinct scopes: the grip area, left/right/bottom joint, and the middle part.

In this stage, the steering wheel’s elliptical shape is transformed into a circle through a geometric homography transformation from the world coordinate system. The subsequent geometric computation addresses several key challenges. One notable challenge is the unknown initial position of the observed object. As a result, 2D homography involves eight degrees of freedom, requiring the calculation of the H matrix (3 ×3) to execute a valid geometric transformation.

The system includes a single RGB camera for capturing video frames and a three-spoke steering wheel (Fig. 5). To conduct experiments, approximately 200 different video frames were captured from various angles to assess the system’s accuracy and performance. The distribution of frames for each variation is as follows: left hand-72, right hand-72, both hands-56. However, the dataset is planned to be expanded later.

Figure 5 RGB images with the corresponding annotations of hand & steering wheel pose.

Results

This work presents a method with a low error margin and establishes a strong baseline for joint pose estimation of two hands interacting with an object, specifically a steering wheel in this case. Table 1 compares techniques for various input modalities for right, left, and two-hand-object pose estimation. The accuracy rates for each case are 33.61%, 52.70%, and 58.92%, respectively.

Figure 6 provides qualitative examples of pose predictions. This approach estimates the pose of two hands interacting with objects from a single RGB image.

In this study, the MediaPipe hand landmark detection pipeline was utilized to track hand movements. Concurrently, the Faster R-CNN architecture was employed to identify the object’s position in the image, resulting in a bounding box around the steering wheel. Subsequently, steering wheel recognition is initiated using the specified three-spoke steering wheel mode. Finally, the position of the hand relative to the steering wheel is calculated: if the hand is within the grip region, the interaction recognition module is activated.

Limitations

While this study demonstrates the feasibility of using a single-camera system to recognize hand-object interactions in driver-steering wheel scenarios, it is important to acknowledge several limitations and opportunities for future work. Despite achieving accuracy rates ranging from 33.61% to 58.92%, further improvements are necessary to ensure robust real-world applications. To enhance generalization to diverse user experiences, it is important to expand the training dataset to encompass a broader range of hand poses, lighting conditions, user characteristics, and steering wheel designs.

Table 1 The impact of various input modalities.

Model	Acc. (%)	
LEFT HAND	33.61	
OBJECT (steering wheel)	48.55	
RIGHT HAND	52.70	
BOTH HANDS	58.92	

Although the current system effectively detects and localizes hands, recognizing and interpreting complex hand gestures for enhanced in-vehicle interactions requires further development. Future research will focus on expanding the vocabulary of gestures.

Discussion

This article introduces a pipeline for recognizing the interaction between a hand and a steering wheel, processing synchronized frames from the video stream. The process involves two main stages: first, using the palm model landmark to track hands, and then estimating the position of the steering wheel based on the model. The proposed method enables real-time detection and localization of the driver’s hands. Currently, the model achieves an accuracy rate of approximately 33.61% for the left hand, 52.7% for the right hand, and 58.92% for both hands. However, future expansion of the dataset will enable further model training and performance improvement, including handling frames with varying lighting conditions. The results demonstrate that this approach ensures hand-object interaction recognition independent of complex and costly installations and devices, making use of simple single cameras.

Figure 6 Qualitative results on the steering wheel dataset.

The approach presented in the article showcases the feasibility and effectiveness of the proposed method, enabling the implementation of natural interactions with a high degree of precision and stability. There are ample opportunities for future research, such as the analysis of driver behavior. Ultimately, this method represents a significant step towards innovative human-vehicle interaction systems, where users can perform gestures and interact with the car without the need for physical manipulation of the steering wheel.

Conclusions

The current state of hand-object recognition is still in the development stage, with new interfaces and human–computer interaction approaches emerging in the market. A prevalent feature among these innovations is the inclination toward reducing user effort and striving to create device-free interactions.

The objective of this work is to develop an affordable and fundamental tool that equips users with essential features for interaction within XR systems. This research has validated the practicality of using this approach for interaction recognition between users and various information systems. A common feature among these systems is the utilization of deep-learning tools. Currently, we possess all the necessary resources to implement frame/video-based action recognition, including mathematical models, datasets, and powerful hardware capable of processing this data for real-time recognition.

In this article, an approach for wheel control through user hand detection was developed. This approach utilized Faster R-CNN and extracted data from a single RGB camera. The authors also curated a dataset comprising 200 samples for model training and evaluation.

Future work will involve expanding the dataset and addressing diverse conditions, such as low lighting and different steering wheel shapes.

Supplemental Information

Supplemental Information 1 Library that provides utility functions for working with COCO datasets

Supplemental Information 2 Engine for the training

Supplemental Information 3 Grouping by images ratio

Supplemental Information 4 DetectionPresetTrain

Supplemental Information 5 Training setting script

Supplemental Information 6 Smoothing labs use

Supplemental Information 7 Evaluation for the COCO

Supplemental Information 8 Matrix transformations

Supplemental Information 9 Annotation for the model

Supplemental Information 10 Detection of the keypoints

Supplemental Information 11 Code and data

Additional Information and Declarations

Competing Interests

Author Contributions

Data Availability

The authors declare there are no competing interests.

Madi Nuralin conceived and designed the experiments, performed the experiments, analyzed the data, performed the computation work, prepared figures and/or tables, authored or reviewed drafts of the article, and approved the final draft.

Yevgeniya Daineko conceived and designed the experiments, performed the experiments, prepared figures and/or tables, authored or reviewed drafts of the article, and approved the final draft.

Shadi Aljawarneh conceived and designed the experiments, performed the experiments, analyzed the data, prepared figures and/or tables, authored or reviewed drafts of the article, and approved the final draft.

Dana Tsoy conceived and designed the experiments, performed the experiments, prepared figures and/or tables, authored or reviewed drafts of the article, and approved the final draft.

Madina Ipalakova conceived and designed the experiments, performed the experiments, analyzed the data, prepared figures and/or tables, authored or reviewed drafts of the article, and approved the final draft.

The following information was supplied regarding data availability:

The raw data and code are available in the Supplemental Files.

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
