# Peer review of "The real-time hand and object recognition for virtual interaction"

_PeerJ Computer Science, doi:10.7717/peerj-cs.2110_

## Round 0.1 · original submission · Major Revisions

Dear authors,
You are advised to critically respond to all comments point by point when preparing a new version of the manuscript and while preparing for the rebuttal letter. Please address all the comments/suggestions provided by the reviewers.

Kind regards,
PCoelho

**Language Note:** The review process has identified that the English language must be improved. PeerJ can provide language editing services - please contact us at [email protected] for pricing (be sure to provide your manuscript number and title). Alternatively, you should make your own arrangements to improve the language quality and provide details in your response letter. – PeerJ Staff

·

Basic reporting

The authors proposed a real-time method for identifying hand-object interactions in extended reality (XR) systems. The proposed system utilized data captured through a single RGB camera during the hand interaction (manipulation scenario) with a steering wheel. The system consists of three components i.e. a hand landmark detector, a steering wheel model tracker, and gesture recognition module designed to analyze interactions between the hand and the steering wheel. The experimental results showed that the interaction between physical objects is realistic in virtual environments with good precision. The research work presented in this manuscript is poor and required the following major changes:
1. Introduction section is poorly written. It lacks the introduction information along with problem statement and objectives of the research work carried out in this manuscript. Rewrite this section carefully.
2. Section “related work” lacks some good research work done in the field of object detection and recognition.

For example “Rautaray, Siddharth S., Real Time Hand Gesture Recognition System for Dynamic Applications (2012). International Journal of UbiComp (IJU), Vol.3, No.1, January 2012”, and “Kim, J.O., Kim, M. and Yoo, K.H., 2013. Real-time hand gesture-based interaction with objects in 3D virtual environments. International Journal of Multimedia and Ubiquitous Engineering, 8(6), pp.339-348.” Please fatten the literature on this topic.

3. Related work section only summarized the existing literature. The critical analysis of these systems are missing. Also the need of this proposed system is not defined in this section.
4. In materials and method section, the overview of the proposed pipeline is given. The title of the manuscript is generalized while this section provide a specific model for hand steering wheel interaction. The proposed system is also required to be generalized for hand-object interaction and this may be used as an example on the generalized model.
5. In algorithm 1, the steps are missing in lines 20-23. Check it and correct.
6. Is the dataset of two hundred (200) images are enough for concluding the results. How much of the image of left hand, right hand and both hands?
7. The authors used Faster R-CNN method. Why? Why not YOLO method. The accuracy of YOLO is more than faster R-CNN. “Ouf, N.S., 2023. Leguminous seeds detection based on convolutional neural networks: Comparison of faster R-CNN and YOLOv4 on a small custom dataset. Artificial Intelligence in Agriculture, 8, pp.30-45.”
8. How the authors measure the natural interaction? What are the parameters for comparing with natural interaction?
9. The paper should be proofread by any professional team as there are many grammatical mistakes in the paper.

Experimental design

Already provided

Validity of the findings

Provided in section 1.

Reviewer 2 ·

Basic reporting

Manuscript Title: The Real-Time Hand and Object Recognition for Virtual Interaction
The manuscript introduces a method for joint pose estimation of two hands interacting with a steering wheel, leveraging MediaPipe hand landmark detection, Faster R-CNN for object position identification, and a 3-spoke Steering Wheel model.
Although the idea seems interesting but there is a lack of innovation and needs to be improved before publication. Here are some suggestions that need to be addressed.
Introduction
Details on current research gaps should be provided.
A smoother flow among topics should be provided.
Specificity in current trends and clarification of objectives should be provided.
More references and citations (related to gesture recognition and information augmentation) are needed to strengthen the academic foundation and credibility of the claims
Related Works
This section provides a comprehensive overview of the existing research landscape related to hand-object interactions in immersive technologies, with a specific emphasis on XR applications.
The lack of critical analysis/discussion of potential limitations in the reviewed studies would add depth and balance.
Discuss research that focuses on reduced/low-cost features-based gesture recognition, such as
• FPSI- Fingertip pose and state-based natural interaction techniques in virtual environment
• Gestures and Marker based low-cost Interactive Writing Board for primary education
• Fingertip Gestures Recognition Using Leap Motion and Camera for Interaction with Virtual Environment
• Two Hand Gesture Based 3D Navigation in Virtual Environments
• Thumb Inclination-Based Manipulation and Exploration, a Machine Learning Based Interaction Technique for Virtual Environments
• Gesture-based guidance for navigation in virtual environment
and others.

Experimental design

Materials & Methods
This section provides a detailed description of the equipment and pipeline used in the study, outlining the approach to recognizing hand-object interactions, specifically focusing on a steering wheel. Several areas could be improved, such as,
Firstly, the rationale behind the choice of equipment, such as the Fanatec Porsche 911 GT2 gaming wheel and Acer C24 monitor, should be explained.
The choice of employing Faster R-CNN for object position identification is noted. However, considering the relatively small dataset of only 200 samples, there are concerns about the model's potential limitations in generalizing to diverse scenarios. Deep learning models, such as Faster R-CNN, typically benefit from larger datasets to capture the variability in real-world conditions. How would you defend it? Why is such a heavy model used?
Recognition module information is limited
Results
The accuracy rates (33.61%, 52.70%, and 58.92%) are presented without comparison to existing literature or benchmarks. Incorporating such comparisons with state-of-the-art methods or discussing the relevance of these accuracy rates within the context of similar studies would enhance result interpretation.
Statistical analysis is insufficient, with the section mentioning a "low error margin" but omitting statistical measures like standard deviation or confidence intervals.
"compares techniques for various input modalities," lacking specifics on the compared techniques. Details on comparison methods, criteria, and statistical significance should be provided.
Individual contributions and impact of MediaPipe hand landmark detection, Faster R-CNN, and 3-spoke Steering Wheel model on results are unclear.
Limitations and challenges encountered during experiments are not addressed, leaving out acknowledgment of potential pitfalls or areas where the proposed method

Validity of the findings

no comment

Additional comments

no comment

Annotated reviews are not available for download in order to protect the identity of reviewers who chose to remain anonymous.
Cite this review as

---

## Round 0.2 · accepted · Accept

Dear authors, we are pleased to verify that you meet the reviewer's valuable feedback to improve your research.

Thank you for considering PeerJ Computer Science and submitting your work.

·

Basic reporting

The authors diligently addressed the first revision's comments, resulting in a refined and polished manuscript.

Experimental design

A well defined experiments are carried out to find the validity of the proposed interaction technique.

Validity of the findings

The findings reported in this article are satisfactory.